# A Retrospective, Single-Center Analysis of Specialized Palliative Care Services for Patients with Advanced Small-Cell Lung Cancer

**DOI:** 10.3390/cancers14204988

**Published:** 2022-10-12

**Authors:** Claudia Wachter, Klaus Hackner, Iris Groissenberger, Franziska Jutz, Lisa Tschurlovich, Nguyen-Son Le, Gudrun Kreye

**Affiliations:** 1Karl Landsteiner University of Health Sciences, Dr. Karl-Dorrek-Straße 30, 3500 Krems, Austria; 2Department of Pneumology, University Hospital Krems, Karl Landsteiner University of Health Sciences, 3500 Krems, Austria; 3Division of Palliative Care, Department of Internal Medicine 2, University Hospital Krems, Karl Landsteiner University of Health Sciences, 3500 Krems, Austria

**Keywords:** palliative care, specialized palliative care, advanced lung cancer, small-cell lung cancer

## Abstract

**Simple Summary:**

Patients with advanced small-cell lung cancer (SCLC) have a considerable symptom burden and may require extensive care. A crucial element of treatment for these patients is the integration of specialized palliative care (SPC). Timely integration of SPC for patients with advanced non-small cell lung cancer (NSCLC) improved quality of life and prolonged survival in large prospective trials. This study provides retrospective data for patients with SCLC with, and without SPC. The results and conclusions indicate that patients with advanced SCLC should participate in a consultation with a SPC team in a timely manner to ensure a benefit of SPC for this patient group.

**Abstract:**

Timely integration of specialized palliative care (SPC) has been shown to improve cancer patients’ quality of life (QoL) and reduced the use of medical services. To evaluate the level of integration of SPC services for patients with advanced small-cell lung cancer (SCLC), we retrospectively analyzed medical records of patients from 2019 to 2021. Regarding the timing of referral to SPC services, we defined four cutoffs for early referral according to the current literature: (a) SPC provided ≤ 60 days after diagnosis; (b) SPC provided ≥ 60 days before death; (c) SPC provided ≥ 30 days before death; and (d) SPC provided ≥ 130 days before death. One hundred and forty-three patients (94.1%) were found to have locally advanced (stage III) or metastatic (stage IV) disease. Sixty-eight were not referred to SPC services (47.6%), whereas 75 patients received SPC (52.4%). We found a significantly higher number of referrals to SPC services for patients with higher ECOG (Eastern Cooperative Oncology Group) (i.e., ECOG ≥ 2) (*p* = 0.010) and patients with stage IV disease (*p* ≤ 0.001). The median overall survival (OS) for SCLC stage III/IV patients (*n* = 143) who did not receive SPC treatment was 17 months (95% CI 8.5–25.5), while those who did receive SPC treatment had a median OS of 8 months (95% CI 6.2–9.8) (*p* = 0.014). However, when we evaluated patients receiving SPC treatment in a timely manner before death as suggested by the different cutoffs indicated in the literature, they lived significantly longer when referred at a minimum of ≥60 or ≥130 days before death. Based on our findings, we suggest that patients with advanced SCLC should participate in a consultation with a SPC team in a timely manner to ensure a benefit of SPC for this patient group.

## 1. Introduction

According to global cancer statistics, lung cancer has become the second most common cancer worldwide (2.21 million cases in 2020) and is also the leading cause of cancer deaths worldwide (1.80 million deaths in 2020) [1]. About 80% of lung cancers are NSCLC, whereas about 15% are SCLC [2]. Although considerable progress has been made in lung cancer therapy throughout the last decade, the five-year survival rate for people with SCLC is around 7%, significantly lower than for those with NSCLC [3].

Novel treatment options and innovative clinical study protocols will lead to increased survival times for patients with advanced lung cancer [4]. Although several progresses has been made, there will remain patients suffering from severe symptoms, including physical and psychosocial issues, requiring SPC [5]. The provision of early palliative care (PC) for patients with advanced lung cancer is recommended by the World Health Organization (WHO) and the American Society of Clinical Oncology (ASCO) [6,7,8].

Of all medical specialties, PC is one of the fastest growing ones. The timely integration of PC is recommended by current guidelines for patients with advanced cancer [9], hence contradicting older models where the transition to PC was performed often in the very last stages of a disease [10]. Nowadays, most international health care services and the WHO recommend referring patients with advanced cancer early in the course of the disease, even when oncologic therapies are still applied [11,12].

As the need for PC for patients with advanced cancers has increased [13], especially because patients are living longer in the era of novel cancer therapeutics [14], timely PC has become more important than early integration, because the sparsity of available resources for SPC must also be considered in this context [9].

General PC is typically provided by physicians and other healthcare professionals from various disciplines [15]. Regarding SPC, there has been growing development in teams providing interdisciplinary and holistic care for patients with advanced cancer [16]. There is evidence from multiple randomized controlled trials that SPC as compared to standard care improves patients’ QoL, symptom control, mood, illness understanding, and end-of-life care [17,18,19,20,21,22,23].

Zimmermann et al. defined SPC as a service delivered by healthcare professionals from at least two different disciplines and who provide or coordinate comprehensive care for patients [24]. Lung cancer patients benefit from SPC, with Temel et al. showing significant improvements in QoL and mood in metastatic NSCLC patients [25]. Furthermore, Temel et al. showed that patients receiving early PC required less aggressive care at the end of life but had longer survival rates compared to patients receiving standard care [25]. However, minimal data are available regarding the role of PC and especially SPC in SCLC [26]. In a large population cohort study of patients with advanced cancer [26], a time frame was chosen based on the literature indicating that PC involvement at least several months before death leads to a benefit for patients with PC needs [27,28]. For several tumor types including SCLC, the point in time that predicted the longest survival time within six months of death was defined as the final cancer-specific transition point [26]. For patients with SCLC, the transition point for the routine integration of PC was 4.2 months (127.75 days) [26]. A review of Davis et al. additionally found several more definitions for “early” PC in patients with serious illnesses in the literature, e.g., the duration of continuity before death, for example ≥30 days or ≥60 days before death [23].

To evaluate the level of integration of patients with SCLC with SPC services at our hospital, we retrospectively analyzed all patients diagnosed with SCLC at a university-based referral center from 2019 to 2021. Additionally, we evaluated different cutoffs for the timepoint of integration of PC according to the literature for SCLC [26], NSCLC [25], and advanced cancer [23].

## 2. Patients and Methods

### 2.1. Study Design

We performed a retrospective analysis of all patients with a diagnosis of SCLC from January 2019 to December 2021.

### 2.2. Setting

#### 2.2.1. Patients

We screened all patients with a diagnosis of lung cancer (International Statistical Classification of Diseases and Related Health Problems (ICD)—Code C34) from January 2019 to December 2021 at a university-based specialized center for lung cancer diagnosis and treatment (University Hospital Krems) and further categorized them according to the histological results of their cancer biopsies. The inclusion criterion was the presence of histologically confirmed SCLC. The exclusion criterion was a patient age of <18 years.

We graded how the disease impacted our SCLC patient’s daily living abilities in terms of their ability to care for themself, daily activity, and physical ability (walking, working, etc.) by means of the Eastern Cooperative Oncology Group (ECOG) performance status at the time of diagnosis. ECOG 0 stands for fully active patients, ECOG 5 stands for dead. For tumor staging, we used the Union Internationale Contre le Cancer (UICC) TNM classification, which is the internationally accepted standard for cancer staging. Follow-up assessments occurred at all clinical visits until death of the patients.

#### 2.2.2. Definition of SPC Services at Our University-Based Referral Center

At our hospital, SPC is provided by PC specialists from at least two different professions (doctor, nurse, psychologist, social worker) that provide or coordinate comprehensive care for patients, as described in the literature [24,29]. In contrast to SPC, general PC is usually provided by physicians and other healthcare professionals from all disciplines at our hospital [15].

The SPC team at the University Hospital Krems is led by interprofessional healthcare professionals including medical doctors, nurses, psychologists, social workers, nutritionists and spiritual care professionals. A clinical ward providing eight beds has approximately 300 admissions per year. Additionally, an outpatient clinic and a PC consultation service provides services for approximately 200 patients per year. The PC consultation service also conducts about 200 home care visits per year. The PC consultation service for both patients in and out of the hospital has approximately 10,000 contacts per year, including multiple visits with the same patient.

#### 2.2.3. Definition of Referral Policy at Our Hospital to the SCP-Team

During the time of this retrospective analysis (1 January 2019–31 December 2021), we had thus far not developed institution-specific consensual criteria as recommended by the literature [9], but the access to SPC was based on a low-threshold approach. Routine screening for PC needs by the oncologist’s triggered referral to the SPC team. Outpatient PC is also available at our institute.

#### 2.2.4. Definition of Transition Points for Referral to SPC

To retrospectively evaluate the timing of referrals to SPC services at our hospital, we defined the following four cutoffs according to the current literature:(a)SPC provided < 60 days after diagnosis, according to Temel et al. [25];(b)SPC provided ≥ 60 days before death, according to Davis et al. [23];(c)SPC provided ≥ 30 days before death, according to Davis et al. [23];(d)SPC provided ≥ 130 days before death, according to Collins et al. [26]

#### 2.2.5. Data Collection

A review of patient records was conducted via an access-limited computer system. Any access to patient records was personalized and monitored. Study-relevant data were pseudonymously compiled and evaluated. Only authorized people had access to the original data.

This retrospective study involving human participants was conducted in accordance with the ethical standards of the institutional and national research committee and with the 1964 Declaration of Helsinki and its later amendments or comparable ethical standards. The study was approved by the local ethics committee (Karl Landsteiner University of Health Sciences Commission for Scientific Integrity und Ethics, No: 1002/2018)

### 2.3. Statistical Methods

The primary endpoint of the study was assessing the prevalence of the integration of PC services. The secondary endpoint of the study was assessing patients’ overall survival (OS), which was defined as the period between the date of SCLC diagnosis and the date of the last follow-up visit or the date of death from any cause. For patients who remained alive, survival times were censored using the date of the last follow-up appointment. To analyze for normal distribution, a Shapiro–Wilk test was used. Chi-square test, Fisher’s exact test, or exact Mann–Whitney U test were used to assess associations between early, late, and no referral to SPC services and clinical parameters. One way ANOVA, Kruskal–Wallis test, Chi-square test, or Fisher’s exact test were used to compare differences among the four defined cutoff-groups for early SPC referral. The Kaplan–Meier estimator method was used to calculate survival probabilities [30]. The log rank test was used to analyze differences between survival curves. To assess the independent effects of co-variables on survival, the cox proportional hazards regression model was used. In case of missing data, which was only present for ECOG values (18% missing at random), we tested whether the proportion of missing data was associated with a different test result. Two-sided tests provided all *p*-values. For all calculations, the Statistical Package for the Social Sciences (SPSS) software, version 28 (SPSS Inc., Armonk, NY, USA), was used.

## 3. Results

### 3.1. Patient Characteristics

Between 1 January 2019, and 31 December 2021, 1260 patients with a newly diagnosed lung cancer were identified at our hospital. Of them, 152 patients had a confirmed diagnosis of SCLC (12%) and were included in the analysis. Further details of patient’s characteristics are described in Table 1.

The distribution of first-line therapy was as follows: surgery in eight (5.3%) cases, radio-chemotherapy in eleven (7.2%) cases, systemic antineoplastic treatment in 112 (73.7%) cases, and best supportive care in 21 (13.8%) cases.

Metastases were present in the following organs: 33 (21.7%) adrenal glands; 32 (21.1%) bone; 46 (30.3%) brain; 46 (30.3%,) liver; 18 (11.8%) lungs; 18 (11.8%) extrathoracic lymph nodes; and 12 (7.9%) pleura. Further characteristics such as UICC stage and ECOG at diagnosis are given in Table 1.

### 3.2. Evaluation of SPC Services for Patients with Locally Advanced or Metastatic SCLC

A total of 143 patients (94.1%) was found to have locally advanced (stage III) or metastatic (stage IV) disease at time of diagnosis of SCLC. We evaluated the prevalence of the utilization of SPC services in these patients: 68 patients (47.6%) were not referred to SPC services, whereas 75 patients (52.4%) received SPC. Furthermore, we evaluated whether there was an association between the referral of patients to SPC services and clinical parameters such as age, gender, smoking status, tumor stage, ECOG performance status, PD-L1 status, first-line therapy, and site of metastases. We found a significantly higher number of referrals to SPC services in patients with higher ECOG (i.e., ECOG ≥ 2) (*p* = 0.010) and in patients with stage IV disease (*p* ≤ 0.001). Furthermore, SPC referrals were made significantly more often for patients with metastases in the adrenal glands (*p* = 0.019) and liver (*p* = 0.048) (Table 2).

We further evaluated the duration of SPC services in days and months. The duration of SPC services was calculated from the day of first contact with the SPC team until the day of last contact (Table 2). Median duration of SPC services (minimum/maximum) was 75 (1–1867) days, or 2.5 (0–62) months (Table 2).

Symptom management (pain 28%; dyspnea 12%; pain and additional other symptoms 5.3%; other symptoms 20%) was the main referral indication for SPC, followed by general referral for integration of PC (26.7%), referral for social aspects (5.3%), and the referral of dying patients (2.7%) (Table 2).

Probability estimates of OS for all patients with SCLC (*n* = 152) were as follows: 28 months (95% confidence interval [CI] 10.4–45.6) for stage II, 20 months (95% CI 12.9–27.1) for stage III, and 8 months (95% CI 6.3–9.6) for stage IV (*p* < 0.003). Three of four patients with UICC stage I at diagnosis were still alive at the end of the study (Figure 1).

### 3.3. OS Related to the Prevalence of SPC Services for Advanced or Metastatic SCLC Patients (Stage III/IV)

At the time of the analysis of the study, 65 patients (86.7%) of the 75 patients who were referred for SPC were already dead. From the 68 patients who were not referred for SPC, 40 patients (58.8%) were dead at the time of analysis. The median OS for SCLC stage III/IV patients (*n* = 143) who did not receive SPC service was 17 months (95% CI 8.5–25.5), while those who received SPC service had a median OS of 8 months (95% CI 6.2–9.8) (*p* = 0.014) (Figure 2).

### 3.4. Referral to SPC Services in Stage III/IV SCLC

We evaluated whether the patients were referred early or late to SPC services. Since early and late referral is not standardized in the literature and guidelines, we used the four different cutoffs as described in the method section (a, b, c, and d) (Figure 3):

(a) Cutoff regarding detection of SCLC and referral to SPC commencing ≤60 days after the first date of detection of locally advanced or metastatic disease [25]:

Regarding this cutoff, 42 of all stage III/IV patients had early integration, and 33 patients had late integration of SPC services. Immediate referral within three days after the first date of detection of advanced disease was present for 14 patients (19%). The log-rank test showed significantly longer survival of patients not referred early as defined by this cutoff (Figure 3). The median OS for patients who were referred early to SPC according to cutoff “a” (≤60 days after diagnosis) was five months (95% CI 1.5–8.4) compared to the median OS for patients who were referred later than 60 days after diagnosis which was 13 months (95% CI 8.9–17.1) (*p* = 0.007).

(b) Cutoff regarding start date of SPC longer than 60 days before death from any cause [23]:

Regarding the second cutoff (“b”), 33 patients were referred early (at a minimum of 60 days before death from any cause), while 32 patients received their referrals within 60 days before their death from any cause. The median OS for patients referred early for SPC according to cutoff “b” (referral earlier than 60 days before death) was 8 months (95% CI 6.4–9.6), while those who were referred within 60 days before death from any cause had a median OS of 3 months (95% CI 0.0–7.2) (*p* = 0.025).

(c) Cutoff regarding start date of SPC longer than 30 days before death from any cause [23]:

Regarding the third cutoff, 40 patients were referred early (at a minimum of 30 days before death from any cause), while 25 patients received their referrals within 30 days before their death from any cause. The median OS for deceased SCLC stage III/IV patients referred for SPC who were referred early (≥30 days before death) according to cutoff “c” was 8 months (95% CI 6.5–9.5), while those who were referred <30 days before death from any cause had a median OS of 6 months (95% CI 0.1–11.9) (*p* = 0.13).

(d) Cutoff regarding start date of SPC longer than 130 days before death from any cause [26]:

Regarding the fourth cutoff, 25 patients were referred early (at a minimum of 130 days before death from any cause), while 40 patients received their referrals within 130 days before their death from any cause. The median OS for deceased SCLC stage III/IV patients referred for SPC who were referred early according to cutoff “d” (referral earlier than 130 days before death) was 9 months (95% CI 4.9–13.0), while those who were referred late was 3 months (95% CI 1.4–4.5) (*p* = 0.003).

Baseline characteristics of the four cutoff groups were then compared (Appendix A) and patients who were referred early to PC according to cutoff “a” were found to have significantly more often ECOG ≥ 2 at time of diagnosis (*p* = 0.002), and best supportive care as the first therapeutic approach (*p* ≤ 0.001) compared to the other three groups. Furthermore, baseline characteristics of early and late referral patients according to all four cutoffs were compared individually. Again, for cutoff a, ECOG ≥ 2 was significantly different between groups (*p* = 0.007). Furthermore, general integration to palliative care was significantly more often a referral indication in all patients who were referred early according to all four cutoffs (Appendix A).

Upon univariate analysis, OS was not associated with age, gender, ECOG performance status, or first-line anticancer therapy but was correlated with referral to SPC services (Table 3).

A multivariate analysis revealed that referral to SPC services was independently associated with OS (HR 0.37 [95% CI 0.19–0.55]; *p* < 0.001) (Table 3).

## 4. Discussion

In the present study, we retrospectively evaluated the referral of patients with SCLC to SPC in a university-based hospital. Next to different patients’ characteristics, indications for SPC, and therapeutic approaches, we also evaluated four different cutoff points to distinguish early vs. late SPC referral. Hence, we found that differences in timepoints of referral to SPC services led to distinct survival outcomes in patients with SCLC. When implementing SPC earlier than 60 days after diagnosis, we observed that the OS for patients with advanced SCLC was significantly shorter. This finding differs to the prospective trial by Temel et al., where patients with metastatic NSCLC who received early PC services lived significantly longer than those who did not receive PC treatment [25]. However, this is not surprising because SCLC differs from NSCLC in terms of illness trajectory [31]. The landmark trial by Temel et al. constituted the concept of early PC, where patients were referred to PC within 60 days after diagnosis [25]. Although the definition of “early PC” has not been established, many trials on early integration used the timeframe of two to three months for their investigations [17,19,21,32,33,34,35]. Nevertheless, a recent analysis found that patients with a higher symptom burden profited more from the PC interventions [36]. Due to a lack of resources, it becomes less possible to provide early access to SPC for all cancer patients with advanced disease [37]. The concept of “timely” PC focuses on referral based on needs [9] and the delivery of services at the optimal time and setting [38]. Recent models for timely and adequate PC suggest offering “targeted PC” only to patients with PC needs instead of treating all patients, providing a more rational use of resources [36]. According to these suggestions, we also evaluated more disease-specific referral points for our patients with SCLC.

According to the literature, most referrals to SPC services occur within 30–60 days before death [23,39]. Therefore, we also included 60 days as one of our cutoff points for early versus late referral in our analysis. When analyzing the time of palliative referral before death, patients with early access to SPC (at least 60 days before death) showed significantly longer survival in our study (*p* = 0.025). We have concluded that those patients who were referred very early (≤60 days) after diagnosis were generally in worse condition, which led to decreased OS. Strengthening this hypothesis is the fact that study patients who were referred for SPC had a significantly worse performance score (ECOG), and that symptom management (pain, dyspnea, other symptoms) was the main reason for SPC referral. Furthermore, we found that these patients significantly more often did not receive an antitumor therapy, but best supportive care only, compared to the patients of the other three groups of early SPC referral cutoffs.

When we compared baseline characteristics of the four cutoff groups, we found that patients who were referred early to PC according to cutoff “a”, the cutoff suggested in the study by Temel et al. [25], had significantly more often ECOG ≥ 2 at time of diagnosis when compared with the other cutoffs suggesting timely referral. An explanation for this observation could be that only patients with a bad performance status were transferred routinely to PC services.

Another finding was that best supportive care as the first therapeutic approach was chosen more often in the early referral group as compared to the other three groups, based on timely referral. This could also indicate that patients who did not receive further anticancer treatment but best supportive care were referred earlier after diagnosis than those receiving anticancer treatment. Similar results were found when baseline characteristics of early and late referral were compared individually for all four cutoffs regarding ECOG performance status. As we only analyzed different cutoffs for early and timely integration in a retrospective manner, it is very difficult to draw conclusions from our study regarding the optimal time point of integration for PC for patients with SCLC. However, as we found a survival benefit for SCLC patients receiving timely integration, this could be the preferred approach for this tumor entity [23,26].

Nevertheless, more recent studies have suggested using a cancer-specific transition point, which would be approximately 130 days for patients with SCLC according to the study by Collins et al. [26]. Our results show a significant difference in survival (*p* = 0.003) if patients are referred to SPC ≥ 130 days before death. Although our results are retrospective, the findings that patients live longer when using the suggested cutoffs from the studies by Collins and Davies [23,26] could hint at the correct time-point of referral for patients with advanced SCLC. Using a correct time-point of referral could be beneficial for patients in terms of increasing their quality of life and preventing suffering [9]. At the end of life, timely access to PC can extend the survival of patients undergoing aggressive treatment. In addition, PC can reduce physical and psychosocial symptom burden [9]. We think that our study results support the concept of timely referral to SPC or PC in patients with SCLC.

Nevertheless, there are still barriers for timely referral to PC [9]. They include a lack of awareness of symptoms and needs, inconsistent referral thresholds, stigma associated with PC, and logistical challenges in referral processes, including limited PC program infrastructure. [9,40,41]. Referral is still often based on the patient’s prognosis rather than symptom control [42].

Hui et al. established a model for providing timely and adequate PC [43]. This model includes four components—namely, (1) the routine screening of supportive care needs at oncology clinics; (2) the establishment of institution-specific consensual criteria for referral; (3) a system in place to trigger a referral when patients meet criteria; and 4) the availability of outpatient PC resources [43]. A limitation of our study is that we had not established institution-specific consensual criteria for referral to SPC services in our hospital. We have recently begun establishing referral criteria for PC patients in our hospital according to the National Comprehensive Cancer Network (NCCN) referral criteria [44], starting with a quality improvement pilot project for patients with advanced cancer in our hospital. We hope that the implementation of the use of referral criteria will lead to timely referral to PC services, hence leading to better outcomes for our patients with advanced cancer.

We were able to show a significant survival benefit if patients were referred earlier than 60 days before death. Access to timely PC can ameliorate the burden of physical and psychosocial symptoms, and it has also been shown to extend the survival of patients undergoing cancer treatment at the end of life [9,45]. Models integrating cancer care and the early integration of PC therapy have been demonstrated to be feasible in phase I and II investigational trials [46]. It has also been found that for patients with metastatic NSCLC, there was not only an improvement in overall survival, but also in quality of life, although patients received less aggressive cancer care and early referral to PC [20]. Regarding SCLC, there are to our knowledge no studies investigating the implementation of PC or SPC thus far.

There are several explanations as to why cancer patients may be reluctant to the timely integration of SPC and may have negative perceptions despite clear benefits. Chosich et al. found that although a majority of patients felt comforted by PC involvement, about 40% felt frightened, and 29% felt hopeless when referred to SPC [47]. The authors of this study concluded that there is still an ongoing need for the better education of patients and the public about PC treatment [47]. There are a few reviews summarizing the findings of trials about the early integration of oncology and PC [11,48]. Nevertheless, there are no guidelines on how to incorporate SPC services or what exactly defines early integration [16]. Nevertheless, a generalizability paradox was found in clinical trials of SPC [49]. The amount and content of oncological consultations or the type or amount of tumor-directed treatment are still not reported in most papers [50].

Definitions of early PC referral may differ between tumor groups. A “one-size-fits all” approach is often used but will not actually “fit all”. For example, the development of bone metastases in the patient with lung cancer confers a substantially different prognosis compared to the same sites of metastases in a patient with breast cancer. Hence, there is a clear need for the testing of cancer-specific time points when referral to PC occurs as “standard quality care” [26]. The authors identified clear disease-specific transition points in the cancer illness that signal a subsequent poor prognosis (<6 months) [51].

A randomized phase II trial examined the feasibility of standardized, early PC (STEP) for patients with advanced cancer [51]. Based on the current international consensus that “early” referral to PC services improves cancer patients’ outcomes, this study addressed the current uncertainty about the best timing of PC integration. Previous work has identified clear disease-specific transition points in the cancer illness that signal a subsequent poor prognosis (<6 months) [51]. The PC protocol developed by Philip et al. should be routinely introduced as a standardized approach (STEP care) for advanced cancer patients and their family caregivers, with referrals at the defined disease-specific, evidence-based transition points [51]. Such studies could serve as a model for future studies at our institution. In our retrospective analysis, we could not evaluate a perfect transition point, as patients with advanced SCLC and early transferal to SPC died earlier.

This study has of course limitations. Our study provides data from a retrospective analysis, limiting the conclusions that can be drawn out of it. Several sources of bias and multiple confounding factors could have led to these findings. Due to these limitations, it is difficult to compare our study to the prospective study performed by Temel et al. [25]. The development of an intervention model for the timely integration of PC care into oncology care is warranted and could be enhanced by further prospective studies, which we strongly suggest since this is only a retrospective analysis.

Another major limitation of our study is that we had no information on QoL for most of the patients. QoL is of major concern for patients with a life-limiting disease such as advanced cancer [25] and is still often not documented in daily clinical routine. When using early referral as a cutoff, we found that patients with worse conditions such as higher ECOG-values and not eligible for further anticancer treatment were referred earlier. Although such parameters are no QoL indicators, they may be surrogate markers for patients whose worse condition may lead to a decrease in QoL.

Results from our retrospective analysis, indicating a lack of documentation of QoL, will lead to compulsory documentation of QoL for advanced cancer patients in our department.

## 5. Conclusions

In our retrospective analysis of patients with advanced SCLC, we found that patients in our study center with early referral after diagnosis showed significant decreased survival when compared to those referred later. We assumed that referral indication of “life-limiting disease”, as recommended by the WHO, is often not the main reason for the integration of PC.

However, symptom control and the burden of symptom control remained the major referral indications for PC. SCLC patients in our study center often showed progressive or advanced disease and benefitted from SPC when referral was provided in a timely, not early manner.

We concluded that the timely integration of PC services, namely the referral indication of “life-limiting disease” in a prospective manner, could improve the quality-of-care and survival for patients with advanced lung cancer. Based on our data, we suggest initiating prospective studies to further resolve the optimal timepoint for PC or SPC in SCLC to improve patients’ quality of life. As advanced SCLC has a very limited prognosis, timely integration of PC or SPC services appears to be a crucial part in the interdisciplinary treatment approach for these patients.

## Figures and Tables

**Figure 1 cancers-14-04988-f001:**
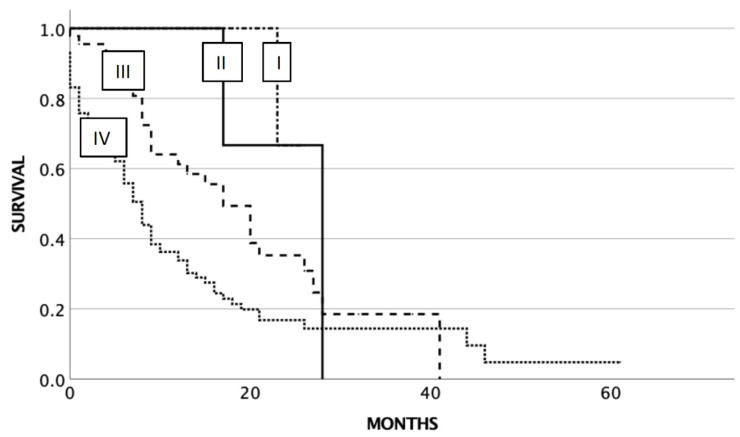
Overall survival related to UICC stage at diagnosis of SCLC.

**Figure 2 cancers-14-04988-f002:**
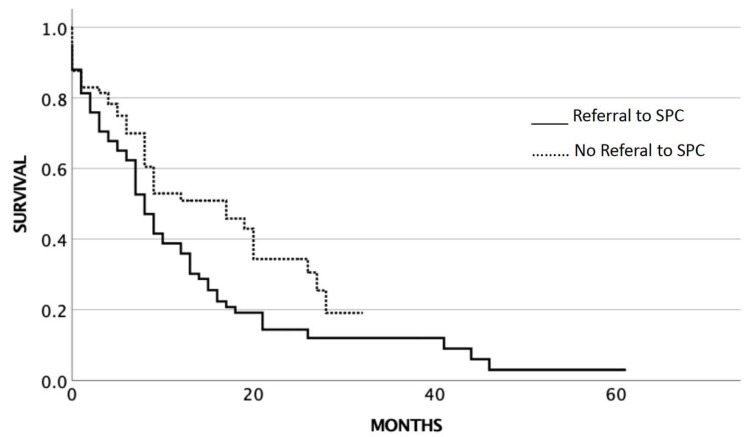
Overall survival related to SPC service in stage III/IV SCLC.

**Figure 3 cancers-14-04988-f003:**
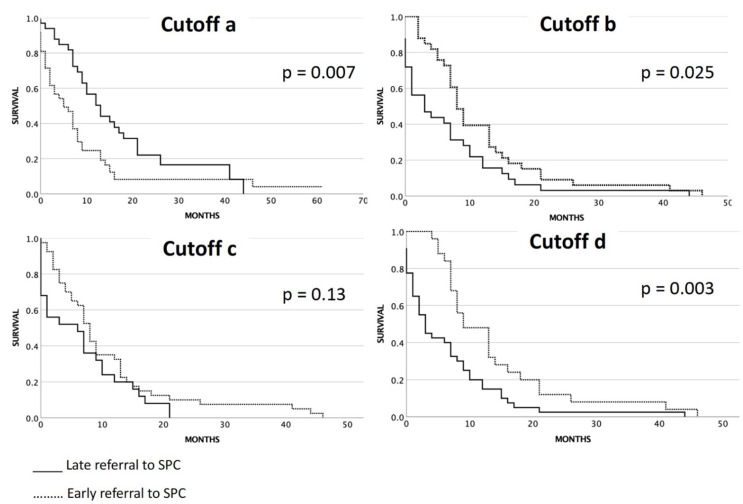
Overall survival related to early or late referral to SPC services with different cutoff values.Cutoff “**a**”: overall survival related to early (≤60 days) or late (>60 days) referral to SPC service after diagnosis; cutoff “**b**”: overall survival related to early (≥60 days) or late (<60 days) referral to SPC service before death from any cause; cutoff “**c**”: overall survival related to early (≥30 days) or late (<30 days) referral to SPC service before death from any cause; and cutoff “**d**”: overall survival related to early (≥130 days) or late (<130 days) referral to SPC service before death from any cause.

**Table 1 cancers-14-04988-t001:** Baseline characteristics of the study patients.

Number of Patients (%)	152 (100)
Age (years), median (range)	68 (37–88)
Male, *n* (%)	90 (59.2)
Female, *n* (%)	62 (40.8)
UICC stage at diagnosis:	
- Stage I, *n* (%)	4 (2.6)
- Stage II, *n* (%)	5 (3.3)
- Stage III, *n* (%)	47 (31.0)
- Stage IV, *n* (%)	96 (63.1)
Metastatic disease at time of diagnosis:	96 (63.1)
- Bone, *n* (%)	32 (21.1)
- Liver, *n* (%)	46 (30.3)
- Lungs, *n* (%)	18 (11.8)
- Pleura, *n* (%)	12 (7.9)
- Brain, *n* (%)	46 (30.3)
- Extrathoracic lymph nodes, *n* (%)	18 (11.8)
- Adrenal glands, *n* (%)	33 (21.7)
History of smoking, *n* (%)	141 (92.8)
Targetable mutation, *n* (%)	0
PD-L1 ≥ 1%, *n* (%)	34 (22.4)
ECOG at time of diagnosis, *n* (%) (*n* = 125):	
- ECOG 0	38 (25.0)
- ECOG 1	69 (45.4)
- ECOG 2	12 (7.9)
- ECOG 3	4 (2.6)
- ECOG 4	2 (1.3)
Therapeutic approaches (first line), *n* (%):	
- Surgery, *n* (%)	8 (5.3)
- Radio-chemotherapy, *n* (%)	11 (7.2)
- Systemic therapy, *n* (%)	112 (73.7)
- Best supportive care, *n* (%)	21 (13.8)
Deaths from any cause, *n* (%)	106 (69.7)
Survival time (months), median (range)	8 (0–61)

UICC = Union Internationale Contre le Cancer, PD-L1 = Programmed death-ligand 1, ECOG = Eastern Cooperative Oncology Group.

**Table 2 cancers-14-04988-t002:** Comparison of stage III/IV SCLC patients (*n* = 143) with and without SPC.

	Referral to SPC Services (*n* = 75)	No Referral to SPC Services (*n* = 68)	*p*-Value
Age (years), median (range)	67 (46–85)	68 (37–80)	n.s. ^†^
Male, *n* (%)	45 (60)	40 (41.2)	n.s. *
Female, *n* (%)	30 (40)	28 (58.8)	n.s. *
UICC stage at diagnosis:			<0.001 *
- Stage III, *n* (%)	12 (16)	36 (52.9)	
- Stage IV, *n* (%)	63 (84)	32 (47.1)	
History of smoking, *n* (%)	72 (96)	62 (91.2)	n.s. *
PD-L1 ≥ 1%, *n* (%)	15 (20)	19 (27.9)	n.s. *
ECOG ≥ 2 at time of diagnosis, *n* (%) (*n* = 116)	14 (22.2)	3 (5.7)	0.010 ^#^
Metastatic disease at time of diagnosis:			
- Bone, *n* (%)	20 (26.7)	12 (17.6)	n.s. *
- Liver, *n* (%)	30 (40)	16 (23.5)	0.048 *
- Lungs, *n* (%)	8 (10.7)	10 (14.7)	n.s.
- Pleura, *n* (%)	9 (12)	3 (4.4)	n.s.
- Brain, *n* (%)	28 (37.3)	18 (26.5)	n.s.
- Extrathoracic lymph nodes, *n* (%)	13 (17.3)	5 (7.4)	n.s.
- Adrenal glands, *n* (%)	23 (30.7)	10 (14.7)	0.019 ^#^
Therapeutic approaches (first line):			n.s. *
- Surgery, *n* (%)	2 (2.7)	1 (1.5)	
- Radio-chemotherapy, *n* (%)	0	2 (2.9)	
- Radiotherapy, *n* (%)	2 (2.7)	5 (7.4)	
- Systemic antineoplastic therapy, *n*(%)	62 (82.7)	48 (70.6)	
- Best supportive care, *n* (%)	9 (12)	12 (17.6)	
Referral indications:			
- Pain, *n* (%)	21 (28)		
- Dyspnea, *n* (%)	9 (12)		
- Other symptoms, *n* (%)	15 (20)		
- Pain and additional other symptoms, *n* (%)	4 (5.3)		
- Dying patient, *n* (%)	2 (2.7)		
- Social aspects, *n* (%)	4 (5.3)		
- General integration of PC, *n* (%)	20 (26.7)		
Duration of SPC services (days), median (range)	75 (1–1876)	-	-
Time from diagnosis till start of SPC (days), median (range)	47 (0–1315)	-	-
Pat referred to SPC services ≤ 60 days after diagnosis, *n* (%) (cutoff a)	42 (56)	-	-
Pat referred to SPC services ≥ 60 days before death, *n* (%) (cutoff b)	33 (44)	-	-
Pat referred to SPC services ≥ 30 days before death, *n* (%) (cutoff c)	40 (53.3)		
Pat referred to SPC services ≥ 130 days before death, *n* (%) (cutoff d)	25 (33.3)		
Deaths from any cause, *n* (%)	65 (86.7)	38 (56.7)	<0.001 *
Survival time (months), median (range)	8 (0–61)	17 (0–32)	0.014 ^‡^

^†^ Mann–Whitney-U test was applied, * chi-square test was applied, ^#^ Fisher’s exact test was applied, ^‡^ log-rank test was applied. UICC = Union Internationale Contre le Cancer, PD-L1 = programmed death-ligand 1, ECOG = Eastern Cooperative Oncology Group, SPC = specialized palliative care, PC = palliative care.

**Table 3 cancers-14-04988-t003:** Univariate and multivariate Cox models.

	Overall Survival	
	Univariate	Multivariate
Variable	HR (95% CI); *p*-Value	HR (95% CI); *p*-Value
Age	0.08 (−0.73–0.23); 0.312	0.05 (−0.12–0.22); 0.552
Gender	0.05 (−0.10–0.20); 0.544	0.06 (−0.11–0.23); 0.472
ECOG ≥ 2	−0.09 (−0.32–0.14); 0.446	−0.19 (−0.43–0.05); 0.124
First-line anticancer therapy	−0.15 (−0.61–0.06); 0.169	−0.03 (−0.25–0.18); 0.768
Referral to SPC services	0.28 (0.14–0.42); <0.001	0.37 (0.19–0.55); <0.001

ECOG = Eastern Cooperative Oncology Group; SPC = Specialized palliative care; HR = Hazard ratio; CI = Confidence interval.

## Data Availability

The data presented in this study are available upon request from the corresponding author. The data are not publicly available.

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
