# Peer review of "A Retrospective, Single-Center Analysis of Specialized Palliative Care Services for Patients with Advanced Small-Cell Lung Cancer"

_cancers, 2022, doi:10.3390/cancers14204988_

Round 1

Reviewer 1 Report

Dear author, I have enjoyed reading the manuscript. The knowledge adds to the understanding of when palliative care is appropriate to initiate. It is also valuable to highlight the aspect of limited resources of specialized palliative care. Possibly this could have been contrasted with the importance of general palliative care at an earlier stage for those who do not have pronounced needs and a large symptom burden.

To further advancing your manuscript I have som parts that you need to consider:

I: Formalities throughout the entire document:

- You write out several abbreviations which you then do not use consistently. This applies to examples: SCLC - small-cell lung cancer // PC - palliative care // SPC - specialized palliative care - These and other abbreviations need to be checked throughout the manuscript and used consistently or not used if the whole words are to be printed.

There are other formal things like double dots in some places..: see for example: line 60, 69, 366; an wrong dot in line 67

I will further point to som aspects in the separate parts:

In the abstract:  the last sentence reflecting the conclusion of the study is very short and reflects only a small part of the result and maybe something about the timing should be included instead.

Introduction: Is clear and informative, ambiguity with abbreviations.

Patients and methods: This part is largely clear. What may help the reader's service is to have included brief explanation about ECOG classification, as well as UICC. 

Results: The first part of the results, which presents patient characteristics and refers to Table 1, contains much of the repetition in the text of what is in the table. Just highlight the main points, then we can read the table. 

Table 1: contains some parts that are not explained: maybe Y should be printed year - a small detail but together with other parts it is not correct. Furthermore, there is no explanation for the Interquartile range - IQR. How this is written within the brackets is also unclear when the reference is (median, IQR) and it is written for example (61-73) put a comma between as you describe it should be. Easy for readers to interpret it as range.

In the table you write best supportive care, while in the text on line 181 you use the term no treatment. Here it needs to be congruent and perhaps we cannot see that best supportive care is no treatment at all. This also concerns Table 2.

Overall, I think the tables could be clearer so that you refer in the way you indicate, e.g. explanation (n, %) and then you write n (%). Se also Survival time in Table 2 (Median, Range) where you have noted median (range).

Figur 1: be clerar that you refer to the figuer in the text. I am unsure if you do despite looking several times.

Figure 2 and 3: You refer to the figures in the headline, isn´t it more correct to refer to them in the text. Tell what the paragraph is about and then refer to the figure in the text.

Line 232: here is not the same sign for number of days as on line 131. Which one should it be?

In lines 236 and 237, distribution is given based on the cutoff for early 39 (52%) referral to palliative care and late integration 36 (48%), respectively. The former is given in Table 2 with the same number, while for the latter, the numbers do not match what is given in the table. The results given in the relevant rows do not refer to Table 2 so should it not be read that way?

Reflect on what is written in the methodology section related to the explanations of the different cutoffs. Now it seems somewhat repetitive. You also use explanations for this in the presentations of the results. Think about this some more so that it is helpful to the reader instead of repetitive.

Discussion: In the discussion you highlight the importance of the study in relation to other studies, its benefits and limitations. All this in a good way it is interesting to read.

On line 323 you write that symptom management for pain, dyspnea and other symptoms was the main reason for referral to specialist palliative care. Where in the results can I read that? The information is interesting, but I miss it in the results.

Author Response

Dear Reviewer,

Thank you very much for taking your time to review our manuscript and adding all those valuable comments! We tried to modify our manuscript according to your suggestions as follows:

Point 1. Formalities throughout the entire document:

We screened our manuscript for abbreviations and used them consistently. We checked for SCLC – small-cell lung cancer, PC-Palliative Care, SPC-specialized palliative care, and others, and consistently used the abbreviation after the first explanation of the abbreviation.

Point 2. Formal things like double dots in some places

We checked the whole manuscript for formal things like double dots in some places and corrected them.

Point 3. Abstract

“The last sentence reflecting the conclusion of the study is very short and reflects only a small part of the result and maybe something about the timing should be included instead.”

We extended the last sentence reflecting the conclusion suggesting that “based on our findings, we suggest that patients with advanced SCLC should participate in a consultation with a SPC team in a timely manner to ensure a benefit of SPC for this patient group”.

Point 3. Introduction

“Ambiguity with abbreviations”

We screened the introduction for abbreviations and used them consistently.

Point 4. Patients and methods

“What may help the reader's service is to have included brief explanation about ECOG classification, as well as UICC.” 

We included a brief explanation about ECOG classification as well as about UICC staging.

Point 5. Results

“The first part of the results, which presents patient characteristics and refers to Table 1, contains much of the repetition in the text of what is in the table. Just highlight the main points, then we can read the table. “

We shortened the description of patient’s characteristics and referred to Table 1. We left the results about anticancer treatment and size of metastases in the text because we think that this information is important for readers.

“Table 1: contains some parts that are not explained: maybe Y should be printed year - a small detail but together with other parts it is not correct. “

We replaced “y” with “years”.

“Furthermore, there is no explanation for the Interquartile range - IQR.”

We replaced “IQR” by “range” according to the suggestion of the reviewer in the next point.

“How this is written within the brackets is also unclear when the reference is (median, IQR) and it is written for example (61-73) put a comma between as you describe it should be. Easy for readers to interpret it as range.”

We changed “IQR” to “range” throughout the manuscript.

“In the table you write best supportive care, while in the text on line 181 you use the term no treatment. Here it needs to be congruent and perhaps we cannot see that best supportive care is no treatment at all. This also concerns Table 2.”

We replaced “no treatment” with “best supportive care”.

“Overall, I think the tables could be clearer so that you refer in the way you indicate, e.g. explanation (n, %) and then you write n (%). See also Survival time in Table 2 (Median, Range) where you have noted median (range).”

We changed this according to the reviewer’s suggestion.

“Figure 1: be clear that you refer to the figure in the text. I am unsure if you do despite looking several times.”

We deleted the sentence “OS analysis for SCLC patients (n = 152) related to stage (Figure 1)” because it was confusing and hope we now fulfilled the reviewer’s suggestion.

“Figure 2 and 3: You refer to the figures in the headline, isn´t it more correct to refer to them in the text. Tell what the paragraph is about and then reference to the figure in the text.”

We changed this according to the reviewer’s suggestion.

“Line 232: here is not the same sign for number of days as on line 131. Which one should it be?”

We corrected this typing error.

“In lines 236 and 237, distribution is given based on the cutoff for early referrals to palliative care and late integration, respectively. The former is given in Table 2 with the same number, while for the latter, the numbers do not match what is given in the table. The results given in the relevant rows do not refer to Table 2 so should it not be read that way?”

We corrected this error.

“Reflect on what is written in the methodology section related to the explanations of the different cutoffs. Now it seems somewhat repetitive. You also use explanations for this in the presentations of the results. Think about this some more so that it is helpful to the reader instead of repetitive.”

We changed the repetitive elements according to the reviewer’s suggestion.

If you have further questions, please do not hesitate to contact us!

Kind regards,

Gudrun Kreye

Reviewer 2 Report

The article “A retrospective, single-center analysis of specialized palliative 2 care services for patients with advanced small-cell lung cancer” retrospectively investigates the prevalence of the integration of palliative care services (primary endpoint) and the question of timely integration of palliative care by comparing the association of early and late referral to palliative care and OS (secondary endpoints) in patients suffering from SCLC.

The primary goal of the study was to evaluate the level of integration of patients with SCLC with SPC services (line 97-98); accordingly, the authors chose as the primary endpoint the “prevalence of integration” and show about 52% of patients received SPC. The secondary endpoint OS was investigated comparing patients with and without SPC referral (table 2 and figure 2). Referral to SPC was associated with decreased OS. This in itself is not surprising nor novel considering the higher amount of patients with UICC stage IV (84 vs. 47%).

The following major methodological and other concerns preclude publication:

Following, the authors evaluate early versus late referral to SPC using different cut-offs described in the literature. Surprisingly, for three out of four cut-offs tested early referral to SPC was associated with increased OS. As the authors state correctly “several sources of bias and multiple confounding factors could have led to these findings (line 388)”. It is very likely, that exactly this is the case. For example, baseline characteristics of early versus late referred patients are not shown for the four cut-offs. Clinically relevant differences in baseline characteristics between the groups could have biased OS. Therefore, descriptive data on baseline characteristics should be shown (especially UICC stage at diagnosis, history of smoking, and PD-L1-status). Along these lines, it is unclear why the authors dichotomized age (“Age>70 years) in their univariate and multivariate Cox models (Table 3).  

In general, it is unclear, why OS was chosen to evaluate referral practice. As palliative care aims to improve quality of life of patients, this outcome would have been better suited to evaluate different referral cut-offs. Even if OS was not confounded by other factors than referral, what would the differences in OS mean? –increasing survival is not a goal of palliative care, but quality of life is.

OS was defined as the period between date of diagnosis and last follow-up visit or death (line 150). The effects of censoring for loss to follow-up in time-to-event analyses are unclear; how many patients (percentage) were lost to follow up? This could have severely biased the results.

Minor Concerns:

1.     Introduction:

·         Line 81, overstatement of survival benefit: only Temel et al. found improved survival, all the other did not; suggest to rephrase.

2.     Methods:

·         Suggest to follow STROBE Statement checklist https://www.equator-network.org/wp-content/uploads/2015/10/STROBE_checklist_v4_cohort.pdf

·         Line 110: “Definition of SPC services at our university-based referral center”: 10,000 contacts a year; does this include multiple visits with the same patient?

·         What was the average intensity of SPC (number of visits? Which professions?)

·         Line 118: What does “counselling team” mean; please explain; is this a palliative care consultation service?

·         Please address any efforts to address potential sources of bias.

·         Please explain how missing data were addressed.

3.     Results:

·         Consider using a flow diagram; include number of patients lost to follow-up.

·         ECOG does not add up to 100%, please explain or mention missing data.

·         Table 1: indicate number of participants with missing data for each variable, indicate statistical tests for each variable (e.g. is missing for “Metastatic disease at time of diagnosis: lung, pleura, brain”).

·         Line 201 and Table 2: Duration of SPC services was calculated, suggest to specify number of visits because this would give a better estimate on how intensively palliative care was conducted.

·         Line 276: 10 days? Typo? Should this be 130 days?

·         Figure 3 legends are not readable; please increase font size that legends can be read when pdf is printed.

·         Line 239: “The log-rank test showed significantly longer survival of patients referred early as defined by this cutoff (p = 0.045) (Figure 3).” Figure 3 shows longer survival for patients which were referred late.

4.     Discussion

·         Sources of bias and possible confounders should be thoroughly discussed. At this point it is difficult to believe that OS is really associated with timely referral.

·         What do the differences in OS mean in regards to palliative care? Survival is not the goal of palliative care; this should be discussed.

·         The authors conclude (line 319) that decreased survival of earlier referred patients (cut-off a) could be caused by patients being in “worse condition”, but no data is shown to support this claim. In addition, how could this be true for cut-off a, but not for the other cut-offs?

5.     Conclusion:

·         Line 400: “SCLC patients typically showed very progressive or advanced disease and benefitted from referral to SPC in terms of prolonged OS when using different cutoffs.” This is not true for cut-off a.

Author Response

Dear Reviewer,

Thank you very much for taking your time to review our manuscript. Our conclusions of the present study are based on retrospective data, therefore we presented them in the conclusion, even though results of a retrospective study cannot be as valuable as prospective data. We regret that we only have retrospective data now. Nevertheless, the results we found support the concept of “timely” referral to palliative care in certain (if not all) types of advanced cancer. Currently, we are developing standard operating procedures based on a model for providing timely and adequate palliative care established by Hui et al.

We tried to answer to all suggestions and concerns as reported below.

We are planning to use the results of this study to plan a prospective study to improve timely integration of specialized palliative care for patients with small-cell lung cancer.

If you have further questions, please do not hesitate to contact us!

Kind regards, Gudrun Kreye

The following major methodological and other concerns preclude publication:

“Following, the authors evaluate early versus late referral to SPC using different cut-offs described in the literature. Surprisingly, for three out of four cut-offs tested early referral to SPC was associated with increased OS. As the authors state correctly “several sources of bias and multiple confounding factors could have led to these findings (line 388)”. It is very likely, that exactly this is the case. For example, baseline characteristics of early versus late referred patients are not shown for the four cut-offs. Clinically relevant differences in baseline characteristics between the groups could have biased OS. Therefore, descriptive data on baseline characteristics should be shown (especially UICC stage at diagnosis, history of smoking, and PD-L1-status).

We included another table to compare clinically relevant differences in baseline characteristics between the four groups (Supplemental Table A). In fact, the reviewer’s suggestion led to interesting findings. When comparing the four groups, group “a” was found to have significantly more patients that were “only” treated by best supportive care. This might have had an influence on the reduced survival of this cutoff-group, since this group did not receive antitumor therapy.

Along these lines, it is unclear why the authors dichotomized age (“Age>70 years) in their univariate and multivariate Cox models (Table 3). “ 

Median age in our study group was 68 years. Therefore, we decided to evaluate, if age >70 or below has in impact on survival by using this as a cut-off in the Cox models. This information is now included in the methods section of the manuscript. However, both age and age>70 did not show a significante regression

In general, it is unclear, why OS was chosen to evaluate referral practice. As palliative care aims to improve quality of life of patients, this outcome would have been better suited to evaluate different referral cut-offs. Even if OS was not confounded by other factors than referral, what would the differences in OS mean? –increasing survival is not a goal of palliative care, but quality of life is.

We agree with the reviewer that quality of life is a main goal of palliative care. However, even the landmark study by Temel et al. reported on survival of NSCLC patients receiving palliative care although this is not the primary objective of palliative care.
Unfortunately, in our retrospective analysis, we had no information on quality of life for most of the patients. This is of course another major limitation of our retrospective analysis.
We stated this major limitation in our discussion.

OS was defined as the period between date of diagnosis and last follow-up visit or death (line 150). The effects of censoring for loss to follow-up in time-to-event analyses are unclear; how many patients (percentage) were lost to follow up? This could have severely biased the results.

We had no patients lost to follow up. Censored patients were still alive; therefore, we used the last date of follow-up visit as the timepoint of censoring.

Minor Concerns:

1.Introduction:

Line 81, overstatement of survival benefit: only Temel et al. found improved survival, all the other did not; suggest to rephrase.

We rephrased this sentence from

 “There is evidence from multiple randomized controlled trials that SPC as compared to primary care not only improves patients’ quality of life, symptom control, mood, illness understanding, and end-of-life care, but also their survival.”

to

“There is evidence from multiple randomized controlled trials that SPC as compared to primary care only improves patients’ quality of life, symptom control, mood, illness understanding, and end-of-life care.”

2. Methods:
Suggest to follow STROBE Statement checklist https://www.equator-network.org/wp-content/uploads/2015/10/STROBE_checklist_v4_cohort.pdf

We modified most of the Methods-Part according to the STROBE Statement checklist.

Line 110: “Definition of SPC services at our university-based referral center”: 10,000 contacts a year; does this include multiple visits with the same patient?

We completed the sentence as follows: The consulting service for both patients in and out of the hospital has approximately 10,000 contacts per year, including multiple visits with the same patient.

What was the average intensity of SPC (number of visits? Which professions?)

The average number of visits by the SPC team is summed up below:

Clinical ward: 300 admissions, outpatient clinic and palliative care consultation service provides services: 200 patients, respectively. The palliative care consultation service also provides about 200 home care visits per year. The palliative care consultation service for both patients in and out of the hospital has approximately 10,000 contacts per year, including multiple visits with the same patient.

We completed this sentence by including the different professions as follows: “The SPC team at the UH Krems is led by interprofessional healthcare professionals including medical doctors, nurses, psychologists, social workers, nutritionists and spiritual care professionals.”

Line 118: What does “counselling team” mean; please explain; is this a palliative care consultation service?

We changed these sentences from: “Additionally, an outpatient clinic and a counseling team provides services for approximately 200 patients per year. The counseling team also provides about 200 home care visits per year.”

to:

“Additionally, an outpatient clinic and a palliative care consultation service provides services for approximately 200 patients per year. The palliative care consultation service also provides about 200 home care visits per year.”

Please address any efforts to address potential sources of bias.

Clinically relevant differences in baseline characteristics between the cutoff groups could have biased OS, therefore we added supplemental analysis on comparison of the different groups in the Supplemental material.

We also emphasized the fact that this is not a comparison study, but a retrospective analysis, and potential biases are inevitable due to the character of the study. However, we think that our results might lead to prospective analysis and may impart new impetus on the topic and the question of the optimal timepoint of the introduction of palliative care in SCLC patients.

Please explain how missing data were addressed.

In case of missing data, which was only present for ECOG values (18% missing at random), we tested whether the proportion of missing data was associated with a different test result.

We included this information in the Method section of the manuscript.

3.Results:

Consider using a flow diagram; include number of patients lost to follow-up.

We had no patients lost to follow up.

ECOG does not add up to 100%, please explain or mention missing data.

We added how we addressed for missing data in the methods section. Missing data was only present for ECOG, and the proportion of missing data did not affect the results.

Table 1: indicate number of participants with missing data for each variable, indicate statistical tests for each variable (e.g., is missing for “Metastatic disease at time of diagnosis: lung, pleura, brain”).

Since patients are undergoing complete staging as part of the standard operating procedure at our hospital (i.e., brain MR for all lung cancer patients; PET/CT or CT of thorax/abdomen plus bone scintigraphy) we feel confident that macroscopic evaluation of metastasis was done properly, without missing data.

Line 201 and Table 2: Duration of SPC services was calculated, suggest specifying number of visits because this would give a better estimate on how intensively palliative care was conducted.

We do agree with the reviewer that this would somehow be of interest. However, in palliative care not every contact to a patient is equal in terms of intensity. Some patients will profit from daily consultations via phone when others require a longer home care visit with application of retarded medication once a week (for example). Both patients benefit from palliative care but cannot be stratified for successful PC in the same manner. Furthermore, we also provide a clinical ward for our patients (approximately 300 admission per year), where inpatient palliative care is applied.

We very much appreciate the reviewer’s suggestion, but we think that providing this information for our study cohort (some patients were supported >800 days) would require a very extensive time for analysis and most probably not add significant value since palliative care is not only assessed by the number of visits.

Line 276: 10 days? Typo? Should this be 130 days?

We corrected this error.

Figure 3 legends are not readable; please increase font size that legends can be read when pdf is printed.

We could provide all four graphs as extra files to the journal for a better resolution and to improve legibility.

Line 239: “The log-rank test showed significantly longer survival of patients erred early as defined by this cutoff (p = 0.045) (Figure 3).” Figure 3 shows longer survival for patients which were referred late.

We corrected this error.

4. Discussion

Sources of bias and possible confounders should be thoroughly discussed. At this point it is difficult to believe that OS is really associated with timely referral.

We tried to include all possible biases in the discussion section. Furthermore, we looked for possible differences within the cutoff groups, as suggested by the reviewer. The retrospective nature of our data is of course limited. Survival was one of other parameters we tried to evaluate along with the implementation of palliative care in SCLC. Nevertheless, we think that our findings may imply further prospective studies for the confirmation of a “timely” integration of palliative care

What do the differences in OS mean in regards to palliative care? Survival is not the goal of palliative care; this should be discussed.

Symptom control and quality of live are undeniable major goals of palliative care. However, patients are often eager to “live”, even in the presence of a chronic disease with inevitable death in the nearby future. We did not want to give the impression that we only focused on “survival” in our study, therefore we tried to rewrite the introduction and discussion session. The main point was to retrospectively evaluate a cohort of SCLC patients with and without palliative care.

The authors conclude (line 319) that decreased survival of earlier referred patients (cut-off a) could be caused by patients being in “worse condition”, but no data is shown to support this claim. In addition, how could this be true for cut-off a, but not for the other cut-offs?

We added additional data (Supplemental Table A) that support the hypothesis that patients of cutoff “a” are in worse condition. They significantly more often did not receive antitumor therapy as first-line but best-supportive care, and they showed a worse ECOG at the time of diagnosis.

However, our data is just retrospectively and therefore vulnerable for biases. We tried to emphasize this in the discussion section and we suggested to initiate further prospective analysis to confirm our findings.

5. Conclusion:

Line 400: “SCLC patients typically showed very progressive or advanced disease and benefitted from referral to SPC in terms of prolonged OS when using different cutoffs.” This is not true for cut-off a.

We adapted this sentence to: “SCLC patients in our study center often showed progressive or advanced disease and benefitted from SPC when referral was provided in a timely, not early manner”.

If you have further questions, please do not hesitate to contact us!

Kind regards,

Gudrun Kreye

Round 2

Reviewer 2 Report

The manuscript has been substantially improved. However, several issues have not been resolved (see below).

The following major methodological and other concerns preclude publication:

“Following, the authors evaluate early versus late referral to SPC using different cut-offs described in the literature. Surprisingly, for three out of four cut-offs tested early referral to SPC was associated with increased OS. As the authors state correctly “several sources of bias and multiple confounding factors could have led to these findings (line 388)”. It is very likely, that exactly this is the case. For example, baseline characteristics of early versus late referred patients are not shown for the four cut-offs. Clinically relevant differences in baseline characteristics between the groups could have biased OS. Therefore, descriptive data on baseline characteristics should be shown (especially UICC stage at diagnosis, history of smoking, and PD-L1-status).

We included another table to compare clinically relevant differences in baseline characteristics between the four groups (Supplemental Table A). In fact, the reviewer’s suggestion led to interesting findings. When comparing the four groups, group “a” was found to have significantly more patients that were “only” treated by best supportive care. This might have had an influence on the reduced survival of this cutoff-group, since this group did not receive antitumor therapy.

Showing the baseline characteristics of the different cut-off populations (Suppl. Table A) added important information e.g. revealing substantial differences between cut-off a and the other cut-offs regarding the ECOG assessment. However, it is unclear why the authors did not provide baseline characteristics of early versus late referred patients (for all of the different cut-offs). The authors claim that OS is significantly different for cut-off a, b, and d (Fig. 3) when early and late referrals are compared. Why do the authors compare the baseline characteristics of four cut-offs (Suppl. Table A) when in fact early versus late referrals are compared in Fig.3?  Showing and evaluating baseline characteristics of early versus late referred patients (for all of the different cut-offs) is important, because it could reveal biases explaining some of the data of Fig.3.

Along these lines, it is unclear why the authors dichotomized age (“Age>70 years) in their univariate and multivariate Cox models (Table 3). “

Median age in our study group was 68 years. Therefore, we decided to evaluate, if age >70 or below has in impact on survival by using this as a cut-off in the Cox models. This information is now included in the methods section of the manuscript. However, both age and age>70 did not show a significante regression.

The authors state that “both age and age>70 did not show a significant regression”. In Table 3, I would suggest to show non-dichotomize age, for the following reasons: Firstly, much information is lost, so the statistical power to detect a relation between the variable and patient outcome is reduced. Indeed, dichotomising a variable at the median reduces power by the same amount as would discarding a third of the data (doi: 10.1136/bmj.332.7549.1080).

In general, it is unclear, why OS was chosen to evaluate referral practice. As palliative care aims to improve quality of life of patients, this outcome would have been better suited to evaluate different referral cut-offs. Even if OS was not confounded by other factors than referral, what would the differences in OS mean? –increasing survival is not a goal of palliative care, but quality of life is.

We agree with the reviewer that quality of life is a main goal of palliative care. However, even the landmark study by Temel et al. reported on survival of NSCLC patients receiving palliative care although this is not the primary objective of palliative care.
Unfortunately, in our retrospective analysis, we had no information on quality of life for most of the patients. This is of course another major limitation of our retrospective analysis. We stated this major limitation in our discussion.

It is understandable that survival is easier to retrieve from standard documentation while assessment of quality of life may not be part of the documentation. The question relevant to this retrospective study is what differences of OS mean in regards to the different cut-offs for early vs. late referral. Would the authors conclude that a positive association between OS and timely referral could hint at the correct time-point for referral? Maybe the authors want to discuss this matter.

OS was defined as the period between date of diagnosis and last follow-up visit or death (line 150). The effects of censoring for loss to follow-up in time-to-event analyses are unclear; how many patients (percentage) were lost to follow up? This could have severely biased the results.

Figure 3 legends are not readable; please increase font size that legends can be read when pdf is printed.

We could provide all four graphs as extra files to the journal for a better resolution and to improve legibility.

I would suggest to change the figure legends by increasing fond size to a degree that the numbers can be identified when the figure is printed in the pdf-fie. This way the figure could be properly interpreted without needing to open a supplementary figure.  

In general, the legends of all figures should be changed from German to English punctuation: 1,0 should be 1.0 and so on.  

The abstract was changed (line 49-51): if understood correctly cut-off a is not associated with longer OS. Why does it say: “However, when we evaluated patients receiving SPC treatment in a timely manner before death as suggested by the four other different cutoffs indicated in the literature, they lived significantly 50 longer.” 

Author Response

Dear Reviewer,

Thank you very much again for taking your time to review our revised manuscript and adding more valuable comments! We tried to modify our manuscript according to your suggestions as follows:

“Showing the baseline characteristics of the different cut-off populations (Suppl. Table A) added important information e.g., revealing substantial differences between cut-off a and the other cut-offs regarding the ECOG assessment. However, it is unclear why the authors did not provide baseline characteristics of early versus late referred patients (for all of the different cut-offs). The authors claim that OS is significantly different for cut-off a, b, and d (Fig. 3) when early and late referrals are compared. Why do the authors compare the baseline characteristics of four cut-offs (Suppl. Table A) when in fact early versus late referrals are compared in Fig.3?  Showing and evaluating baseline characteristics of early versus late referred patients (for all of the different cut-offs) is important, because it could reveal biases explaining some of the data of Fig.3. “

We calculated and now provide information on the baseline characteristics of all four cutoffs (Suppl.Tables B-E). In fact, the additional data showed that for patients referred according to cutoff a, ECOG ≥ 2 was (again) significantly different between the groups. On the other side, the indication for patients who were referred early according to cutoff b and c was significantly more often a “general integration to palliative care”, which would further contribute to the concept of “timely” referral. However, also patients who were referred early according to cutoff d had significant more often a general integration of palliative care as indication for referral.
We thank the reviewer for pointing to this additional data analysis. The tables are provided as supplemental material and they are shortly described in the results section.

The authors state that “both age and age>70 did not show a significant regression”. In Table 3, I would suggest to show non-dichotomize age, for the following reasons: Firstly, much information is lost, so the statistical power to detect a relation between the variable and patient outcome is reduced. Indeed, dichotomising a variable at the median reduces power by the same amount as would discarding a third of the data (doi: 10.1136/bmj.332.7549.1080).

We removed the dichotomized age-calculations and now show non-dichotomized age throughout the manuscript according to the reviewer’s suggestion.

It is understandable that survival is easier to retrieve from standard documentation while assessment of quality of life may not be part of the documentation. The question relevant to this retrospective study is what differences of OS mean in regards to the different cut-offs for early vs. late referral. Would the authors conclude that a positive association between OS and timely referral could hint at the correct time-point for referral? Maybe the authors want to discuss this matter.

Thank you for this valuable discussion point! We discussed this association between OS and timely referral in the discussion part regarding a possible positive association between OS and timely referral hinting at the correct time-point for referral.

I would suggest to change the figure legends by increasing fond size to a degree that the numbers can be identified when the figure is printed in the pdf-fie. This way the figure could be properly interpreted without needing to open a supplementary figure.

We increased the font size of all figures (and the graphical abstract) and hope that this allows a proper interpretation of the charts now.

In general, the legends of all figures should be changed from German to English punctuation: 1,0 should be 1.0 and so on.  

We changed the legends of all figures: the decimals are now consistently dots instead of commas.

The abstract was changed (line 49-51): if understood correctly cut-off a is not associated with longer OS. Why does it say: “However, when we evaluated patients receiving SPC treatment in a timely manner before death as suggested by the four other different cutoffs indicated in the literature, they lived significantly 50 longer.” 

We thank the reviewer for this important observation. We corrected the abstract (line 64-65), since only cutoff b (early = ≥ 60 days before death, suggested by Davis et al.) and d (early = ≥ 130 days before death, suggested by Collins et al.) showed significantly longer survival for early referrals.

If you have further questions, we would be glad to answer them.

Round 3

Reviewer 2 Report

The following minor concerns preclude publication:

1.     Comparison of baseline characteristics for all cut-offs were introduced as Supp. Tables B-D. I think this improved the manuscript because possible major biases can now be excluded. However, the median survival time (last line of Supp. Table C (cut-off b) and D (cut-off c)) seem to be mixed up with Supp. Table B (cut-off a): at the moment the numbers are the same for all three cut-offs 5 vs 12 months OS.

For cut-off a 5 vs. 13 months OS are described in the text (ll. 355) while Supp. Table B states 5 vs. 12 months. Is this correct?

2.     Non-dichotomize age is now shown in Table 3. The methods section needs to be changed accordingly (ll. 233).

Author Response

Dear Reviewer,

Thank you very much again for taking your time to review our revised manuscript and adding more valuable comments! We tried to modify our manuscript according to your suggestions as follows:

  1. Comparison of baseline characteristics for all cut-offs were introduced as Supp. Tables B-D. I think this improved the manuscript because possible major biases can now be excluded. However, the median survival time (last line of Supp. Table C (cut-off b) and D (cut-off c)) seem to be mixed up with Supp. Table B (cut-off a): at the moment the numbers are the same for all three cut-offs 5 vs 12 months OS. For cut-off a 5 vs. 13 months OS are described in the text (ll. 355) while Supp. Table B states 5 vs. 12 months. Is this correct?

We corrected the survival time of Supp. Table C and Supp. Table D, since it was mixed up with Supp. Table B. The survival time is now correctly given in all tables.

  1. Non-dichotomize age is now shown in Table 3. The methods section needs to be changed accordingly (ll. 233).

The methods section was adapted.